# Memory-electroluminescence for multiple action-potentials combination in bio-inspired afferent nerves

Kun Wang[1,5], Yitao Liao[1,5], Wenhao Li[1,5], Junlong Li[1], Hao Su[1], Rong Chen[2], Jae Hyeon Park[3], Yongai Zhang[1,2], Xiongtu Zhou[1,2], Chaoxing Wu [1,2] ✉, Zhiqiang Liu [4] ✉, Tailiang Guo [1,2] ✉ & Tae Whan Kim [3] ✉

The development of optoelectronics mimicking the functions of the biological nervous system is important to artificial intelligence. This work demonstrates an optoelectronic, artificial, afferent-nerve strategy based on memory-electroluminescence spikes, which can realize multiple action-potentials combination through a single optical channel. The memory-electroluminescence spikes have diverse morphologies due to their history-dependent characteristics and can be used to encode distributed sensor signals. As the key to successful functioning of the optoelectronic, artificial afferent nerve, a driving mode for light-emitting diodes, namely, the non-carrier injection mode, is proposed, allowing it to drive nanoscale light-emitting diodes to generate a memory-electroluminescence spikes that has multiple sub-peaks. Moreover, multiplexing of the spikes can be obtained by using optical signals with different wavelengths, allowing for a large signal bandwidth, and the multiple action-potentials transmission process in afferent nerves can be demonstrated. Finally, sensor-position recognition with the bio-inspired afferent nerve is developed and shown to have a high recognition accuracy of 98.88%. This work demonstrates a strategy for mimicking biological afferent nerves and offers insights into the construction of artificial perception systems.

Bio-inspired electronic systems with artificial functions, such as biosensing[1-3], bionic robotics[4-6], and neuromorphic computing[7-10], have attracted much interest. Especially, the realization of distributed, parallel, and event-driven information input and processing is of importance for bio-inspired electronic systems[11]. When an external stimulus affects the receptors in the human nervous system, it generates potential changes. These changes are then combined into a single afferent fiber. The cerebral cortex processes and recognizes the signals generated by the irritation and provides feedback in the human body[12], as shown in Fig. 1a. Similar to the human nervous system, scientist have an urgent desire to find ways to transmit the excitation signals from distributed sensors via artificial afferent nerves in bio-inspired electronic systems[13]. Artificial electronics, basic unit of electronic systems with artificial intelligence, are of great significance for the establishment of artificial nervous systems. However, in typical electronic systems, the electrical signals from distributed sensors

[1]College of Physics and Information Engineering, Fuzhou University, Fuzhou 350108, China. [2]Fujian Science & Technology Innovation Laboratory for Optoelectronic Information of China, Fuzhou 350108, China. [3]Department of Electronic and Computer Engineering, Hanyang University, Seoul 133-791, Korea. [4]Research and Development Center for Semiconductor Lighting Technology, Institute of Semiconductors, Chinese Academy of Sciences, Beijing 100083, China. [5]These authors contributed equally: Kun Wang, Yitao Liao, Wenhao Li. ✉e-mail: chaoxing_wu@fzu.edu.cn; lzq@semi.ac.cn; gtl_fzu@hotmail.com; twk@hanyang.ac.kr

**Fig. 1 | Artificial brain recognition based on the afferent nerve. a** Process of human brain recognition for external stimulation. Potential changes of the receptors are generated by external stimulation and are combined into a single afferent fiber to generate action potentials. The brain processes the input information to achieve position judgments. **b** Artificial brain recognition system based on the artificial afferent nerve. When the sensors are triggered, electrical signals are generated to drive Nano-LEDs to emit memory-electroluminescence (Mem-EL) spikes. The Mem-EL spikes transferred in a single light fiber are transmitted to a convolutional neural network for recognition, mimicking the brain's processing and recognition. **c** Schematic of the Nano-LED, which has an ITO/ $Al_2O_3$ /nano-LED/ $Al_2O_3$ /ITO structure. **d** Transmission electron microscope image of the Nano-LED.

should be encoded as voltage pulse sequences to map changes in the external excitation, which is totally different from the efficient spike-based signal transmission in the human nervous system[14].

External excitations, for instance, pressure, electricity, light, etc., are usually used to generate the corresponding electrical or optical feedback from the electronics to mimic the function of the biological system[15–20]. Compared to electrical signals, optical signals have the advantages of high speed, large bandwidth, and spatial transmission signals[21–24]. Therefore, a promising approach would be to use optical pulse signals in an artificial nervous system. Researchers have developed a series of neuromorphic optoelectronic devices, such as two-terminal optical synapses[25–27] and three-terminal optoelectronic transistors[28–30]. However, current optoelectronic devices typically use light signals as input stimuli or generate light signals of varying intensity in response to electrical stimulation. It is difficult to apply these optoelectronic devices to electronic system with multiple action-potentials because how to achieve the encoding of light signals is challenging. Therefore, optical-encoding schemes for artificial nervous systems must be further developed.

In this work, we demonstrate that a nanoscale light-emitting diode (Nano-LED) operating in the non-carrier injection mode with memory-electroluminescence (Mem-EL) behavior can be used to transfer electrical signals to history-dependent optical signals. Those multiple optical signals can then be transmitted into a single light fiber. The history-dependent luminescence characteristic is defined as that the current luminescence state is highly dependent on the luminescence history. Therefore, as long as the EL intensity of the previous moment is different, the light signal is different even though the amplitude of the currently applied voltage is the same. In other words, the device is capable of memorizing the luminescent state of the previous moment. The Mem-EL with interesting history-dependent characteristics is the result of electron oscillations in the multiple quantum wells (MQWs) of the Nano-LEDs. The Mem-EL spikes triggered by distributed sensors are received and transmitted to a convolutional neural network for recognition, which mimics the brain's recognition for spike signals (Fig. 1b). In addition, we demonstrate that wavelength-division multiplexing of light by using Nano-LEDs with different wavelengths can be used to increase the bandwidth of transmitted information.

## Results
### Design and performance of Nano-LEDs
As shown in Fig. 1c, the Nano-LED contains gallium nitride (GaN)-based nanorod LEDs sandwiched between two aluminum oxide ($Al_2O_3$) insulating layers. The $Al_2O_3$ insulating layers can block charge carrier

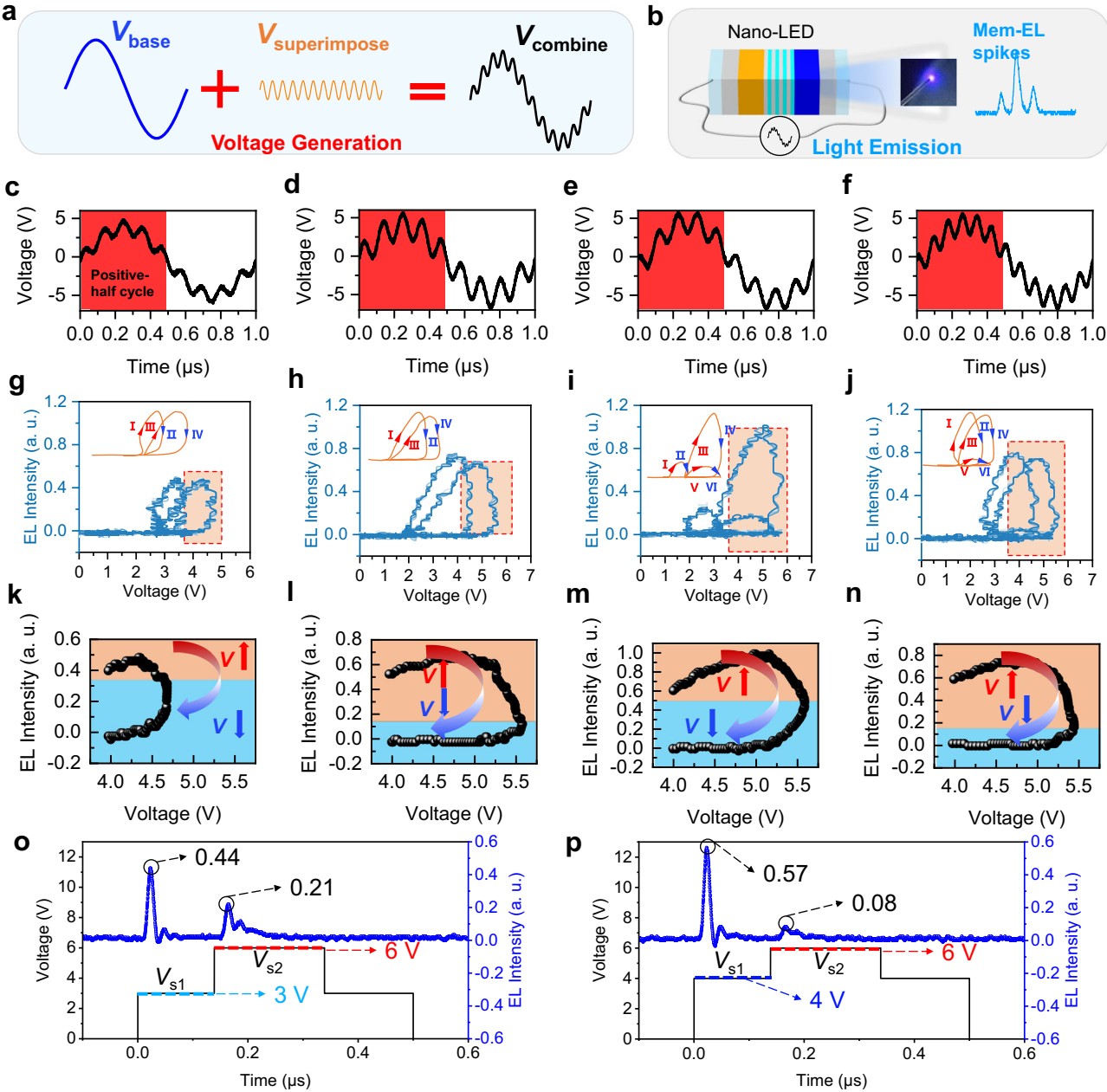

**Fig. 2 | Characteristics of the Nano-LED. a** Generation of $V_{combine}$. $V_{combine}$ can be obtained by superimposing $V_{superimpose}$ on $V_{base}$. **b** Light emitted from a Nano-LED driven by $V_{combine}$. **c**–**f** Different $V_{combine}$ used to drive a Nano-LED by adjusting the amplitude, phase and frequency of $V_{superimpose}$. The EL can only be observed in the positive-half cycle of $V_{base}$ (red area). **g**–**j** EL intensity-voltage relationship to the variation of $V_{combine}$ during the positive half-cycle of $V_{base}$. The inset shows the trend of the EL intensity with voltage, where the blue arrows represent decreasing EL intensity and the red arrows represent increasing EL intensity. **k**–**n** Effect of a rising and then falling voltage near the peak of $V_{combine}$ on the EL intensity. The red arrows represent increasing voltage and the blue arrows represent decreasing voltage. **o** EL spikes generated by a Nano-LED driven by two consecutive square signals ($V_{s1} = 3$ V and $V_{s2} = 6$ V) and (**p**) by another two consecutive square signals ($V_{s1} = 4$ V and $V_{s2} = 6$ V).

injection from the external indium-tin-oxide (ITO) electrodes (Supplementary Figs. 1, 2). Therefore, under high-frequency alternating-current AC voltages, the electrons inside the LED will oscillate periodically in the MQWs (Fig. 1c), leading to Mem-EL light emission. Figure 1d shows the details of the device structure, where an $Al_2O_3$ insulating layer (60 nm) and an ITO electrode (100 nm) are deposited sequentially on the LED. The sapphire substrate and the ITO layer on the transparent glass are used as the bottom insulating layer and the bottom electrode, respectively. That the introduction of the insulating layer is the key to generating Mem-EL spikes, which will be discussed in detail later, is worth noting.

A combined voltage signal ($V_{combine}$) is used to generate Mem-EL spikes (Fig. 2a) and consists of two parts: (1) The first is a base sinusoidal voltage ($V_{base}$) with a single-period sinusoidal signal. (2) The second is a superimposed sinusoidal voltage ($V_{superimpose}$) with a smaller amplitude and a higher frequency. As shown in Fig. 2b, a periodic radiative recombination of the carriers occurs inside the Nano-LED under $V_{combine}$ and generates the Mem-EL spikes. Interestingly, the morphology of the Mem-EL spikes can be varied by adjusting the amplitude ($A$), frequency ($f$), and phase difference ($\Phi$) of $V_{superimpose}$ while $V_{base}$ remains unchanged (Supplementary Fig. 3). Figure 2c–n show the variations in the properties of the Mem-EL under different

$V_{combine}$. A typical waveform for the $V_{combine}$ of $V_{superimpose}$ superimposed on $V_{base}$ is shown in Fig. 2c–f. The EL intensity-voltage relationships under different $A, f$, and $\Phi$ are also presented to demonstrate the Mem-EL process more clearly. Here, the applied voltage increases and decreases periodically (inset of Fig. 2g–j). As illustrated in Fig. 2g–j, two or even more different EL intensities can be observed under the same voltage, which is quite similar to the widely reported behavior of a memristive device. The EL intensity-voltage relationship shows a hysteresis behavior (Fig. 2k–n).

Moreover, the hysteresis behavior of the Mem-EL spikes will differ depending on the values of $A, f$, and $\Phi$. No linear correlation exists between the EL intensity and the applied voltage amplitude, which is utterly different from the behavior of a conventional LED in the ordinary operation mode (Supplementary Movies 1–3). As is well known, for a memristive device, the current-voltage curve shows a hysteresis-loop characteristic[31–33]. Although the memory characteristics of the proposed Nano-LED are different from those of memristive devices, they have similar output performances. To illustrate history-dependent luminescence or memory electroluminescence more clearly, we further provide the EL intensity-voltage characteristics driven by AC voltages with different amplitudes (Supplementary Fig. 4). The lower the applied drive voltage is, the lower the brightness and the smaller the opening of the hysteresis loop are. As the amplitude is increased, the EL intensity increases, which produces a larger hysteresis-loop opening. Therefore, the hysteretic EL intensity-voltage curve is similar to the hysteretic current-voltage curve in memristive devices that have history-dependent characteristics. Therefore, we tend to say our device has history-dependent luminescence.

To further demonstrate this history-dependent luminescence characteristic, we used two consecutive square signals (the voltages are defined as $V_{s1}$ and $V_{s2}$) to drive the Nano-LED, as shown in Fig. 2o, p. When $V_{s1} = 3$ V and $V_{s2} = 6$ V, the first EL intensity is 0.44, and the second EL intensity is 0.21. However, when $V_{s1}$ is increased to 4 V and $V_{s2}$ is kept constant at 6 V, the first EL intensity increases to 0.57, and the second EL intensity decreases to 0.08. Therefore, the current EL intensity is not necessarily determined by the current voltage but is influenced by the previous EL intensity. In other words, the current EL intensity greatly depends on the historical EL intensity, which is the history-dependent luminescence characteristic of the device.

## Operating mechanism of Mem-EL

The finite element analysis method is used to study the operation mechanism of the Nano-LED (Supplementary Table 1), and the structure of the simulated Nano-LED is shown in Fig. 3a. After considering simulation convenience and computational complexity, we set the thickness of the $Al_2O_3$ to be 100 nm in the simulation. Even though the thickness of the sapphire insulating layer will have an influence on the EL intensity, the purpose of the simulations is to demonstrate the history-dependent luminescence properties of the devices. Therefore, this simulation mode can demonstrate the mechanism behind the generation of history-dependent luminescence. $V_{combine}$ is applied to the established Nano-LED model (Supplementary Fig. 5), and as is well known, the region of the MQWs contributes to generating luminescence from GaN-based LEDs[34,35]. Therefore, studying the concentration distribution of carriers near that region helps in the understanding of the Mem-EL characteristics. The distributions of the electron and the hole concentrations ($C_e$ and $C_h$) in the MQWs during a period of $V_{combine}$ are presented in Fig. 3b, c. The experimental and simulation results show that the luminescence occurs only in the positive half-cycle of the drive signal, which is determined by the structure of the pn-junction. An applied electric field acting on the pn-junction is generated under the positive half-cycle of the AC drive signal. When this applied field is larger than the LED turn-on threshold, the carriers inside the Nano-LED are driven into the MQWs to generate radiative recombination, which is equivalent to the forward bias of the pn-junction. However, due to the existence of the insulator, the EL will stop because there are no externally injected carriers. The electrons/holes are accumulated at the p-GaN/insulator interface and the n-GaN/insulator interface, respectively. When driven by the negative half-cycle of the AC drive signal, a reverse electric field acting on the pn-junction is generated. The reverse electric field, together with the built-in field will release the charge accumulated by the previous positive half-cycle voltage, allowing the internal carriers of the LED to be restored to their initial states or sufficient carriers to be accumulated for the radiative recombination in the next positive half-cycle (Supplementary Fig. 6 and Supplementary Note 1).

Driven by the positive-half cycle voltage, holes from p-GaN and electrons from n-GaN are transported into the MQWs, and radiative recombination occurs (Supplementary Figs. 7, 8). Under the periodic increase and decrease of the applied voltage, radiative recombination does not occur at once but in batches, resulting in Mem-EL spikes with multi-peaks (Supplementary Figs. 9, 10). As shown in Fig. 3b, due to the relatively high mobility of electrons, the trend of the electron concentration in the MQWs is consistent with that of $V_{combine}$. In contrast, the concentration distribution of the holes shows multiple oscillations over time. This is similar to changes in the morphology of the Mem-EL spikes, which indicates that holes largely determine the formation of Mem-EL spikes, as shown in Fig. 3c. Of note is that when two consecutive voltage peaks with the same amplitude ($V_1$ and $V_2$ in Fig. 3c) are applied, the variations in hole concentration in the MQWs are different, and these variations cause the EL intensities to be different. More specifically, the hole concentration under $V_1$ is more significant than that under $V_2$, which means the hole concentration for current-state electroluminescence is determined by the applied voltage and influenced by the EL history.

The average electron and hole concentration in the MQWs provides a clearer demonstration of Mem-EL, as depicted in Fig. 3d. When driven by $V_{combine}$, the electron concentration remains high while the hole concentration exhibits a significant variation similar to the EL waveform. If the working mechanism of Nano-LEDs is to be explored, not only the carrier concentration in the MQWs, but also the electron and hole concentrations at the n-GaN/insulator interface and p-GaN/insulator interface, respectively, must be considered[36–39]. As illustrated in Fig. 3e, f, n-GaN and p-GaN exhibit a depleted state, resulting from the depletion of carriers without external replenishment[40–42]. Moreover, the depletion states at both ends are voltage dependent, and the depletion region of p-GaN is larger than that of n-GaN. The length of the depletion region mainly depends on the doping concentration of GaN (Supplementary Fig. 11). In the simulation model, the doping concentration of holes in the p-region is smaller than the doping concentration of electrons in the n-region (Supplementary Table 1). Therefore, the p-region, which has a low hole concentration, requires a longer depletion region to provide an equivalent number of holes as electrons.

According to the above results, a carrier transport model of the Nano-LED with Mem-EL characteristics is proposed and is shown in Fig. 4. Note that the change rate of the voltage signal will lead to changes in the operating state of the device because of the non-carrier injection mode. Even when a positive voltage is applied to the device, the current in the external circuit is reversed as long as the voltage is on the falling edge. Similarly, when the voltage is on the rising edge of the negative half cycle, the current in the external circuit is positive, which is completely different from the conventional DC mode of operation.

As shown in Fig. 4a, the total number of electrons or holes used for radiative recombination in a $V_{combine}$ cycle is defined as $Q$, and only the carrier transport process under a voltage pulse is considered. The Nano-LED is in thermal equilibrium when no voltage is applied. Once a voltage pulse is applied and starts to increase ($V_1$), radiative recombination of the holes from p-GaN with the electrons from n-GaN occurs in the MQWs (Fig. 4b). If the number of electrons or holes consumed at

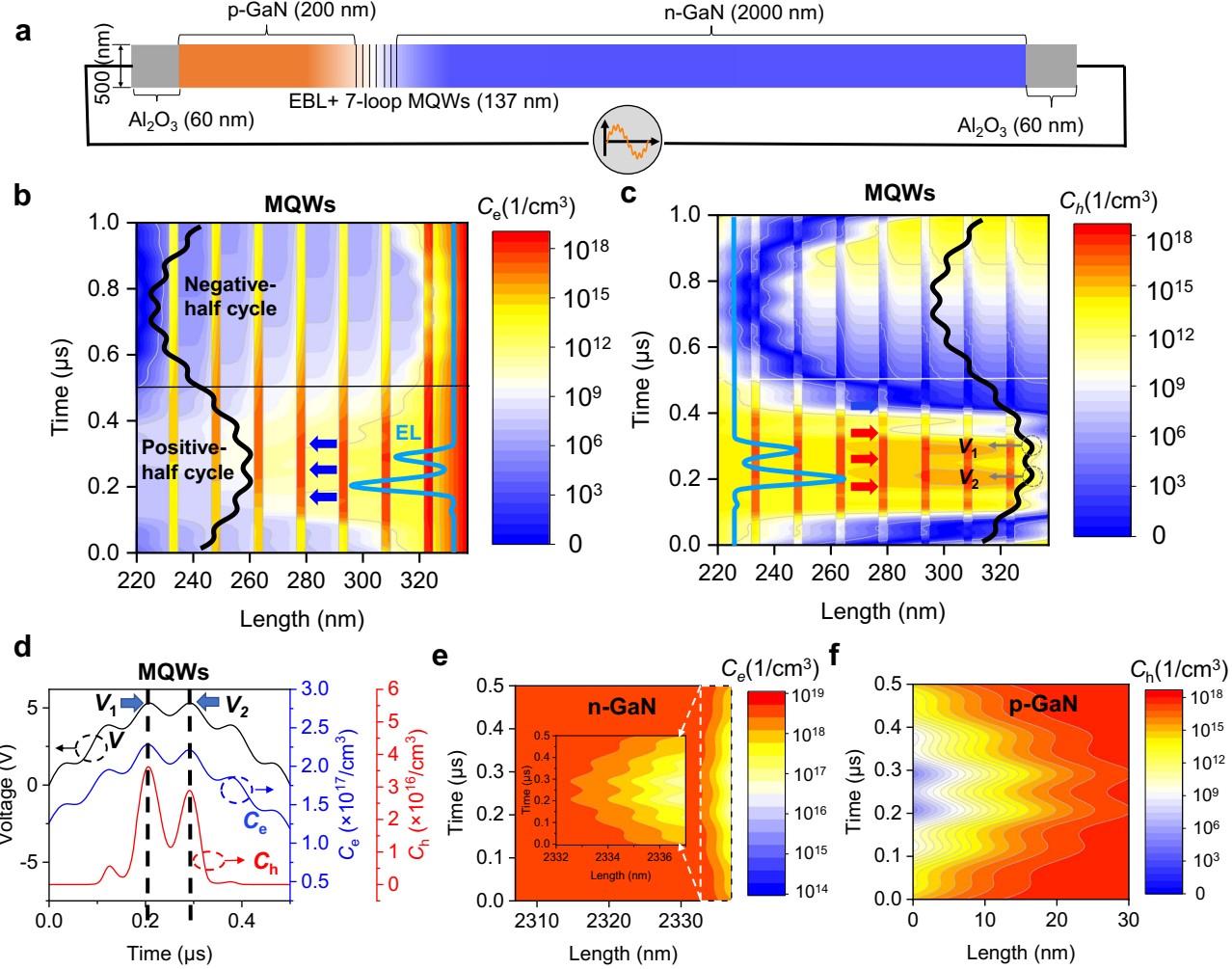

**Fig. 3 | Finite element analysis (FEA) for the Mem-EL. a** Structure of the simulated device. **b** Electron and (**c**) hole concentration ($C_e$ and $C_h$) redistributions in the MQWs during one drive cycle. The black line is the driving voltage $V_{combine}$, and the blue line is the EL waveform. The blue arrows represent the direction of electron motion, the red arrows represent the direction of hole motion. **d** Average electron (blue line) and hole (red line) concentrations in the MQWs under $V_{combine}$ (black line). $V_1$ and $V_2$ represent two consecutive voltage peaks with the same amplitude, respectively. **e** Electron concentration redistribution in n-GaN. **f** Hole concentration redistribution in p-GaN.

this stage is assumed to be $Q_1$, the number of remaining holes or electrons is $Q - Q_1$. When the voltage reaches its maximum, the remaining carriers are used for further radiative recombination ($Q_2$), as shown in Fig. 4c. Due to the insulating layer, external carriers cannot be injected into the Nano-LED to replenish the consumed carriers, leading to the formation of a depletion region at the GaN/insulator interface. The induced electric field caused by the depletion region would prevent further radiative recombination.

When the voltage decreases (at this point the applied voltage is $V_2$, and $V_2 = V_1$), the induced electric field is greater than the applied electric field. Therefore, the carriers accumulated at the electrode/insulator interface decrease, leading electrons and holes to move to the n-GaN and the p-GaN, respectively. However, the luminescence does not stop immediately because the remaining carriers in the MQWs can still be used for radiation recombination (Fig. 4d). Therefore, the number of carriers ($Q_3$) used for radiation recombination is smaller than $Q_1$, and the EL intensity is smaller than that in Fig. 4b, leading to the hysteretic EL intensity-voltage loop. Thus, the behavior of the EL intensity with time shows a dependence on the history of the EL.

## Performance of the bio-inspired afferent nerve

According to the above analysis, the state of the EL intensity at the present time is influenced by historical electrical stimulations, which greatly influence Mem-EL spikes. In other words, the Nano-LED can generate different Mem-EL spikes under different $V_{combine}$. Thus, we can utilize this electro-optical conversion process to mimic the generation of multiple action-potentials and their combination in bio-inspired afferent nerves. Here, a tactile perception process in an artificial neural system based on the Nano-LED is demonstrated, including the sensor trigger, voltage signal generation, signal driving, light signal generation, reception, and recognition (Supplementary Fig. 12). For the realization of sensor encoding and voltage signal generation, an 8-bit digital signal encodes 256 sensors, as shown in Fig. 5a. For example, the sensor 'S104' is encoded as 01101000 based on decimal-to-binary encoding rules. Here, the first 2 bits '01', the middle 4 bits '1010', and the last 2 bits '00' are mapped to the amplitude ($A = 1\,V$), frequency ($f = 12\,MHz$), and phase difference ($\Phi = 0°$) of $V_{superimpose}$, respectively (Supplementary Table 2). Finally, the combined signal $V_{combine}$ can be obtained by superimposing $V_{superimpose}$ on $V_{base}$ (Supplementary Fig. 13). Notably, the $V_{base}$ used in this work is fixed, with an amplitude of 5 V and a frequency of 1 MHz (Supplementary Fig. 14).

Different combinations of $A$, $f$, and $\Phi$ will lead to 256 different $V_{combine}$. As shown in Fig. 5b, when $f$ and $\Phi$ are not changed ($f = 9\,MHz$ and $\Phi = 0°$), four different $V_{combine}$ can be obtained with different values of $A$ (0.5 V, 1 V, 1.5 V, 2 V). Furthermore, when $A$ and $\Phi$ are not changed ($A = 1\,V$ and $\Phi = 0°$), another four different $V_{combine}$ can be

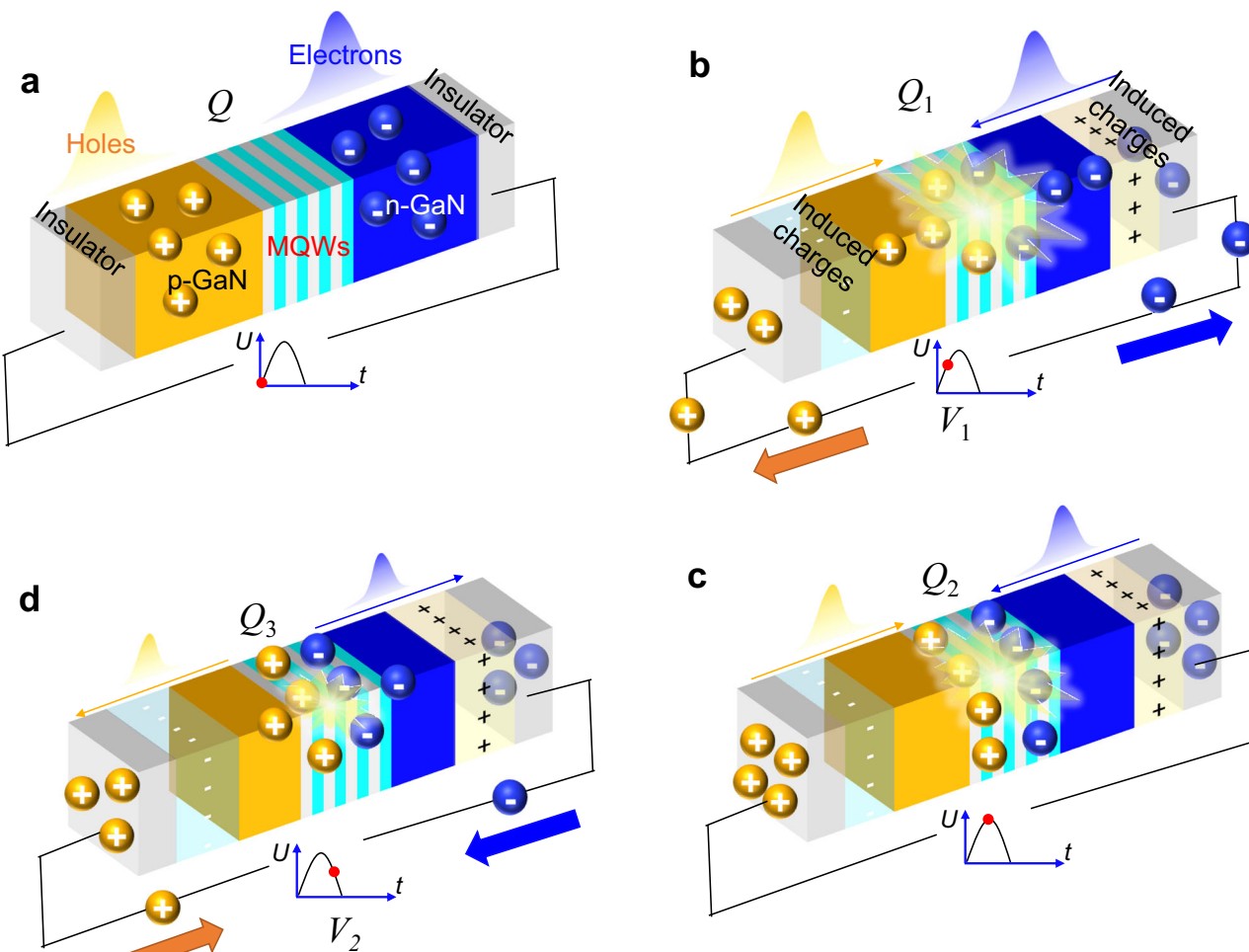

**Fig. 4 | Carrier transport model of the Nano-LED. a** Initial state of the device. The number of holes or electrons for radiative recombination in a $V_{combine}$ period is defined as $Q$. **b** Radiative recombinations of holes and electrons during increases in $V_1$. The number of electrons or holes consumed at this stage is defined as $Q_1$. **c** Further radiative recombination occurs when the voltage reaches its maximum value. The number of electrons or holes consumed at this stage is defined as $Q_2$. **d** Final radiative recombination when the voltage decreases to $V_2$ ($V_2 = V_1$). The number of electrons or holes consumed at this stage is defined as $Q_3$ ($Q_3 < Q_1$), $Q = Q_1 + Q_2 + Q_3$.

obtained with different values of $f$ (9 MHz, 10 MHz, 11 MHz, 12 MHz), as shown in Fig. 5c. Similarly, four different $V_{combine}$ can be obtained by changing only $\Phi$ ($A = 1$ V, $f = 9$ MHz and $\Phi = 0°$, $90°$, $180°$, $270°$), as shown in Fig. 5d. Therefore, this encoding and mapping process converts sensor triggers to $V_{combine}$. Driven by these conditions of $V_{combine}$, the Nano-LED generates various history-dependent Mem-EL spikes, as shown in Fig. 5e–g.

In order to make the variation in the waveforms of the Mem-EL spikes more apparent, we present the variation of the EL intensity with $A$, $f$, and $\Phi$ in Fig. 5h–j. When $f$ and $\Phi$ are fixed and only $A$ is changed, the number and position of EL spikes remains almost unchanged, but the EL intensity changes and increases with increasing $A$ (Fig. 5h and Supplementary Fig. 15). On the other hand, when $A$ and $\Phi$ are fixed, the number, position and EL intensity change significantly with increasing $f$ (Fig. 5i). Similarly, the number, position, and EL intensity change when only $\Phi$ is changed (Fig. 5j). The rich variation of Mem-EL spikes has promising applications in mimicking afferent nerve function in biological nervous systems.

Compared to electrical signals, optical signals have the advantage of being able to realize wavelength-division multiplexing (WDM). Here, we demonstrate two-channel WDM, which was realized using two Nano-LEDs (blue and green), as shown in Fig. 6a. The center wavelengths of the two devices are 451.4 nm and 519.8 nm, respectively

(Supplementary Fig. 16). To ensure that as much light as possible can pass through the filters, we chose two filters with center wavelengths of 450 nm and 520 nm, respectively. Worth noting is that, in practical applications, the light signals pass through the filters before training and recognition. Although the light is attenuated as it passes through the filter, as long as enough light passes through the filter, the recognition of the light signal will not be affected. Therefore, sufficient light passes through the filters due to their having 88% transparency, so the light signal is sufficient for training and recognition, and does not affect the experimental process.

When two different $V_{combine}$ are applied to the blue and green Nano-LEDs, Mem-EL spikes with different wavelengths and different morphologies are generated. We assume that the $V_{combine}$ applied to blue Nano-LEDs is encoded as '11010111' while the $V_{combine}$ applied to green Nano-LEDs is encoded as '01001100'. The two optical signals with different wavelengths are multiplexed into a single optical fiber for transmission, generating a signal that is composed of Mem-EL spikes and carries two encoded spikes. At the receiving end, filters of specific wavelengths can be used to filter out the blue and green signals, and the Mem-EL spikes are decoded to obtain the hidden information.

Figure 6b, c show the combined EL spikes, the blue EL spikes and the green EL spikes driven by four different $V_{combine}$. The first $V_{combine}$

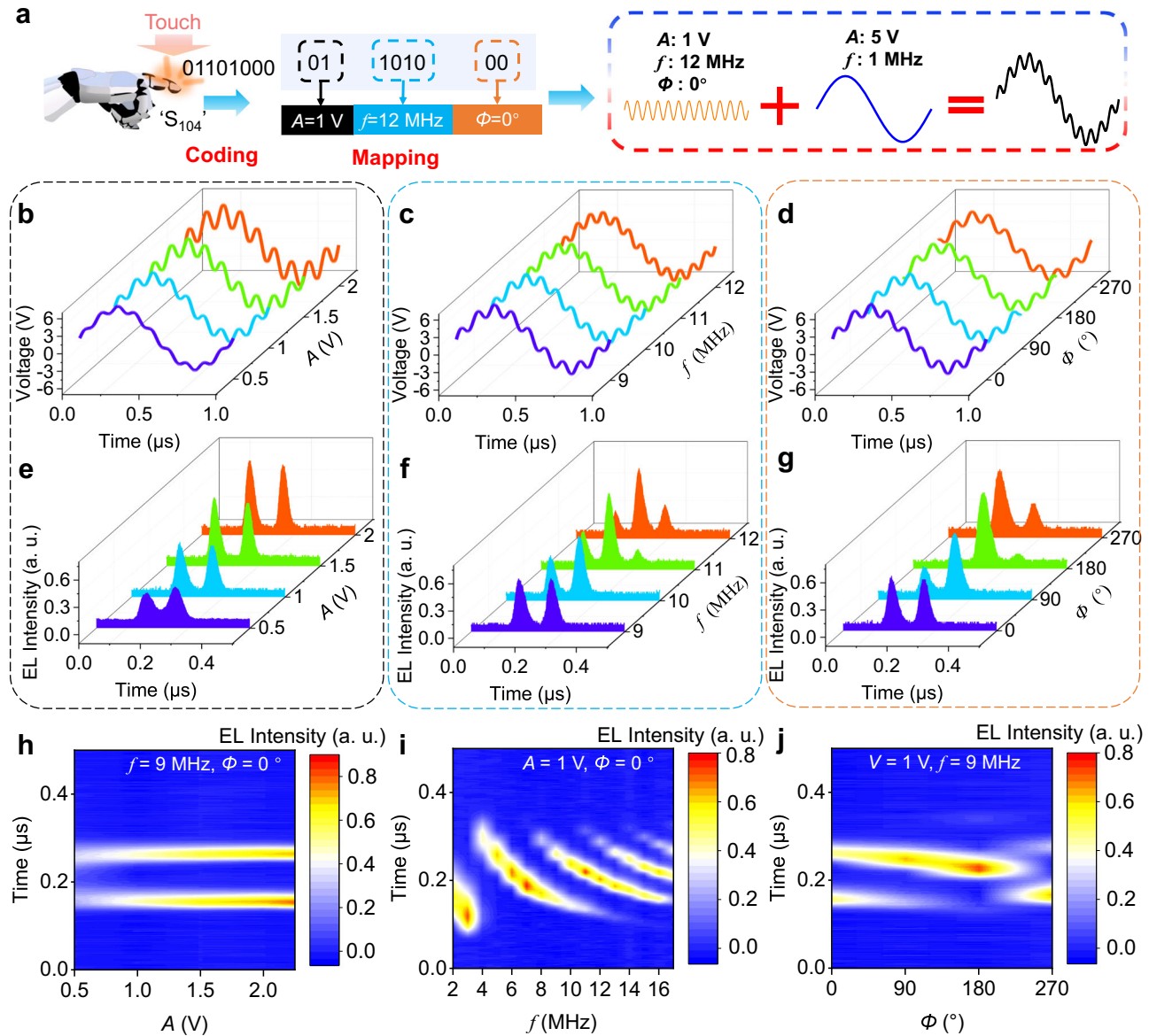

**Fig. 5 | Encoding of artificial receptors, generation of $V_{combine}$, and the corresponding typical Mem-EL spikes. a** An example of encoding for sensor 'S104' and mapping to $V_{combine}$. Each sensor is encoded as an 8-bit binary code, where the first 2 bits represent the amplitude of $V_{superimpose}$, the middle 4 bits represent the frequency of $V_{superimpose}$, and the last 2 bits represent the phase of $V_{superimpose}$. $V_{combine}$ is obtained by superimposing $V_{base}$ on the 8-bit mapped $V_{superimpose}$. **b–d** $V_{combine}$ that vary with amplitude, frequency, and phase. **e–g** EL spike waveform that varies with amplitude, frequency, and phase. **h–j** Heat maps of the EL intensity that varies with amplitude, frequency, and phase.

encoded as '11001000' and the second $V_{combine}$ encoded as '10101000' are applied to the green Nano-LEDs and blue Nano-LEDs, respectively. The waveform of the generated green spike (Fig. 6b) is superimposed on the waveform of the generated blue spike (Fig. 6c) to obtain a cyan spike with a waveform carrying the hidden information (Fig. 6d). Similarly, when driven by another two $V_{combine}$ that are encoded as '11010111' and '10000110', by superimposing the waveform of the blue spike (Fig. 6e) on the waveform of the green spike (Fig. 6f), a cyan spike with a waveform (Fig. 6g) carrying other hidden information is obtained.

Artificial intelligence is used to recognize these Mem-EL spikes to judge the triggered artificial sensors, which can mimic the human brain's response to tactile perception, as shown in Fig. 7a. This study designed a convolutional neural network based on a gramian angular field (GAF) residual network (ResNet) for recognizing EL spike waveforms[43–45]. The input Mem-EL spike waveform is a one-dimensional sequence. The GAF is used to convert this one-dimensional sequence into two-dimensional matrix data to enhance the features of the Mem-EL spike's waveform. Moreover, a residual network is used for further feature extraction, and 256 waveform features are obtained after several convolution and pooling calculations. Finally, the obtained features are input to the fully connected layer for 256 classification outputs (Supplementary Fig. 17 and Fig. 7b).

The well-trained neural network can accurately recognize 256 Mem-EL spike waveforms, in which almost 100% recognition accuracy is achieved on the training and validation sets. The testing accuracy is as high as 98.88% (Fig. 7c). Finally, software platforms are employed to achieve the sensor-position recognition required for bio-inspired afferent nerves, as shown in Fig. 7d. The eight artificial receptors in the virtual robot's palm are defined as 'S47', 'S102', 'S121', 'S92', 'S119', 'S34', 'S49', and 'S56', respectively. When these receptors are touched, the signal generator is triggered to generate a corresponding voltage signal that drives the Nano-LED to emit Mem-EL spikes. These Mem-EL spikes are received by the APD

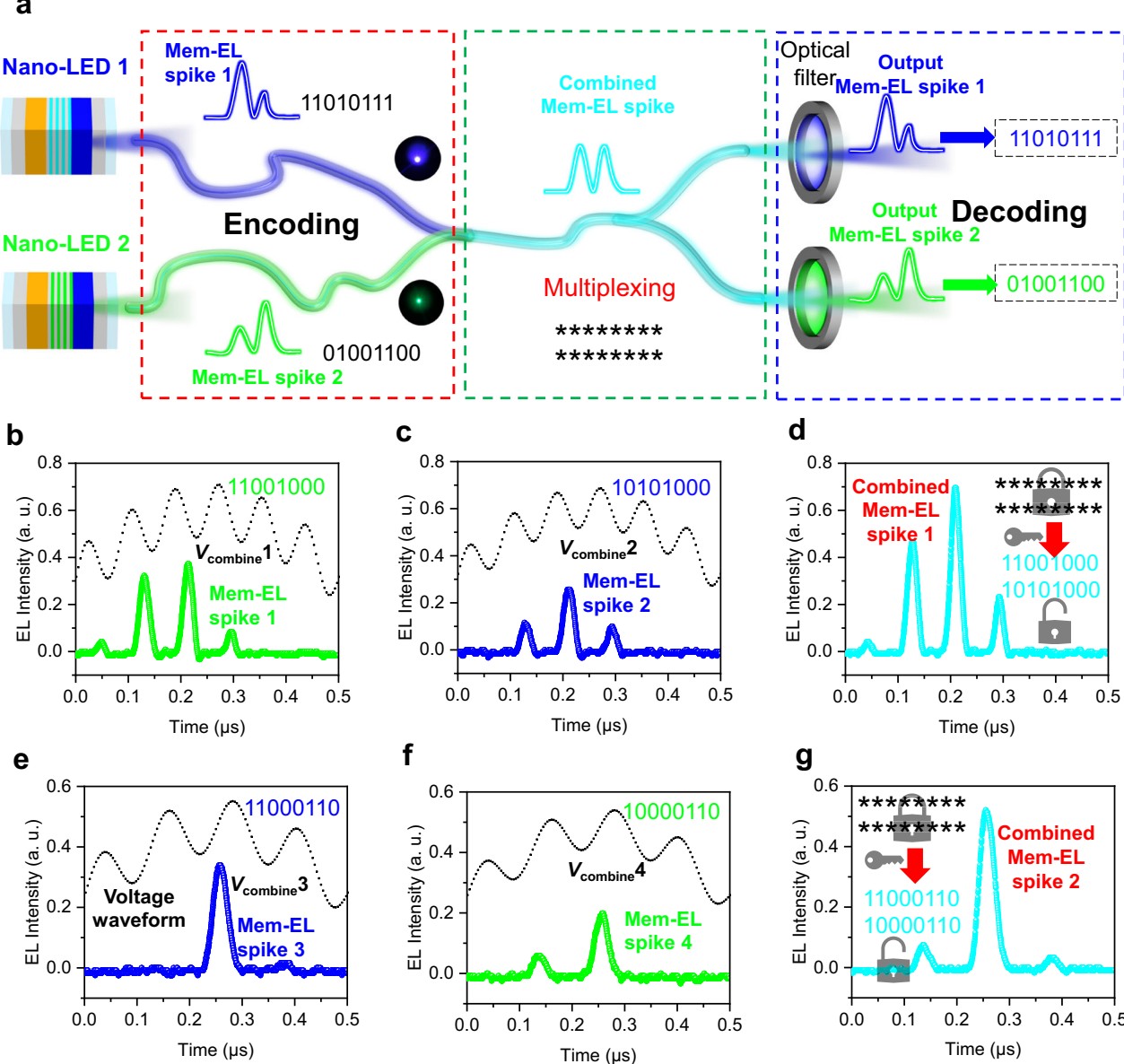

**Fig. 6 | Demonstration of wavelength-division multiplexing (WDM) based on blue and green Nano-LEDs. a** Schematic diagram of the two-channel WDM. When the $V_{combine}$ is applied to the blue and the green Nano-LEDs, two kinds of Mem-EL spikes carrying different information are generated. The two optical signals are multiplexed to form a combined Mem-EL spike. At the receiving end, optical filters with a specific wavelength are used to obtain Mem-EL spikes carrying the original information. **b, c** Mem-EL spikes generated by Nano-LEDs driven by two $V_{combine}$ coded as (**b**) '11001000' and (**c**) '10101000', respectively. **d** Combined Mem-EL spike carrying the hidden information. **e, f** Mem-EL spikes generated by Nano-LEDs driven by two $V_{combine}$ coded as (**e**) '11000110' and (**f**) '10000110', respectively. **g** Combined Mem-EL spike carrying other hidden information.

and read into the well-trained residual neural network based on the Gramian angular field (GAF-ResNet) for recognition (Supplementary Fig. 18 and Supplementary Movie 4). In this process, Nano-LEDs combined with artificial intelligence technology are used to mimic the combinations of multiple action-potentials in bio-inspired afferent nerves, which is expected to promote the development of optoelectronic devices for artificial nervous systems.

## Discussion
This work demonstrates a nanoscale, optoelectronic device based on a Nano-LED with history-dependent luminescence. The conversion of electrical signals to memory-electroluminescence (Mem-EL) spikes is achieved using this device, which mimics the generation of multiple action-potentials and their combinations in bio-inspired afferent nerves. Moreover, the software and hardware platforms are combined

to build sensor-position recognition using the bio-inspired afferent nerves. Finally, a GAF-ResNet is designed to recognize, with a high recognition accuracy of 98.88%, these Mem-EL spikes to mimic the human brain's response to tactile perceptions. This study provides an idea for mimicking artificial, biological perception systems, which has great application prospects in building electronic devices for artificial nervous systems.

## Methods
### Fabrication of the Nano-LEDs
$SiO_2$ nanospheres (600 nm in diameter) are used as nano-masks. Inductively-coupled-plasma etching is used to etch the LED wafer. A 60 nm-thick layer of $Al_2O_3$ is deposited as an upper insulating layer on the Nano-LED by using magnetron sputtering. A 100 nm-thick ITO electrode is deposited on the upper insulating layer by using

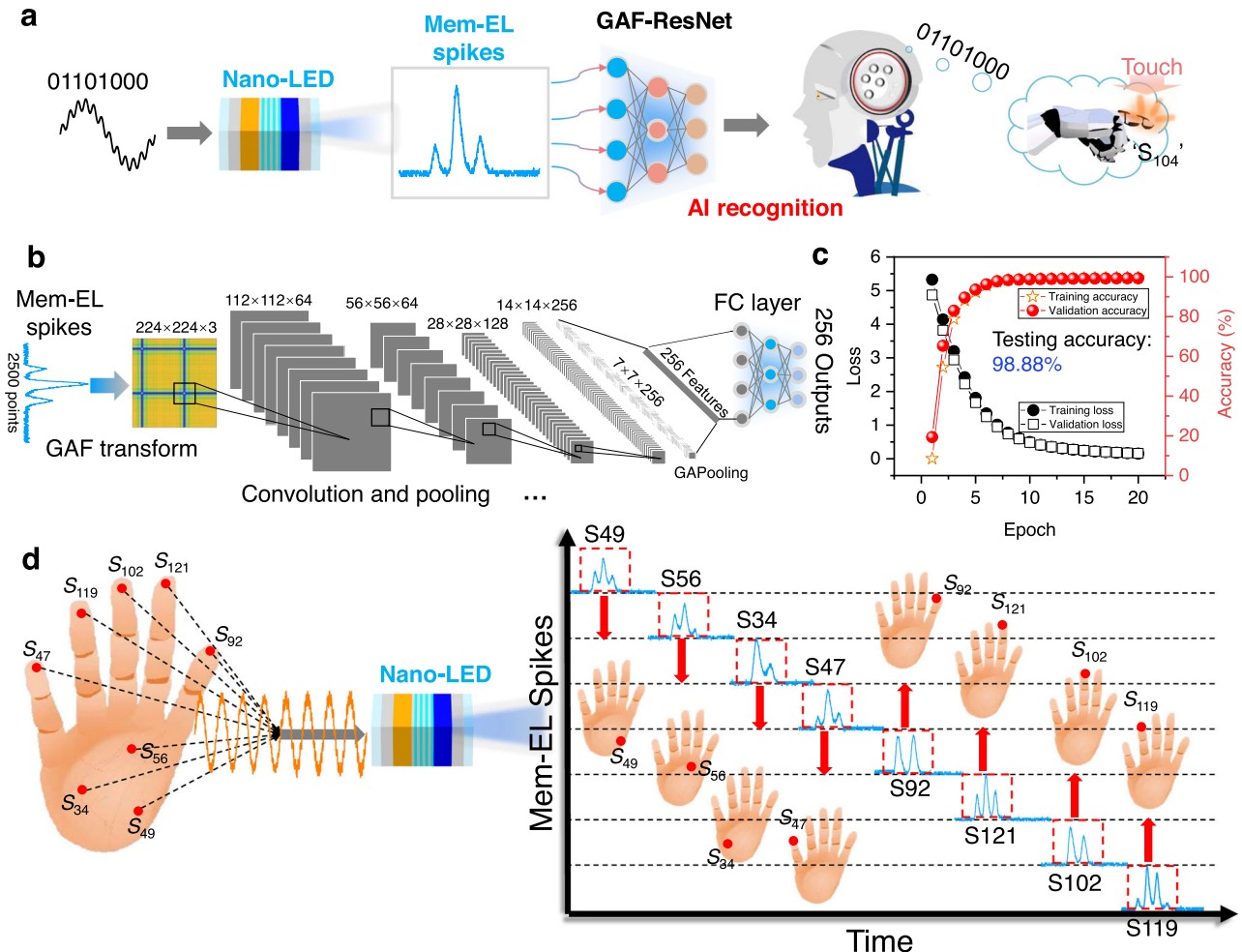

**Fig. 7 | Sensor-position recognition system based on this Nano-LED. a** An example of recognition of sensor 'S104'. $V_{combine}$ is used to drive the Nano-LED to generate Mem-EL spikes, and an artificial neural network is used to recognize the Mem-EL spikes for the judgment of the sensor being touched. **b** Structure of the residual neural network based on the Gramian angular field (GAF-ResNet) for Mem-EL spike recognition. **c** Accuracy and loss of recognition for Mem-EL spikes by GAF-ResNet. **d** Schematic diagram of the recognition process of multiple sensors being triggered based on the Nano-LED.

magnetron sputtering. A sapphire substrate (650 μm in thickness) is used as the bottom insulating layer, and an ITO film on transparent glass is used as the bottom electrode.

## Simulation

The finite element method is used to simulate the Nano-LED with a structure of $Al_2O_3$ (60 nm)/p-GaN (200 nm)/AlGaN electron blocking layer (20 nm)/a 7-period QW (well length: 3 nm, barrier length: 12 nm) /n-GaN (2000 nm)/$Al_2O_3$ (60 nm). The detailed simulation parameters are shown in Supplementary Table 1, and the $V_{combine}$ used in the simulation of Nano-LED is shown in Supplementary Fig. 5.

## Measurements

The $V_{combine}$ is compiled using a computer and generated using a function generator (RIGOL, DG5352). The electrical characteristics and light waveform are recorded using an oscilloscope (RIGOL, DS2302A). The EL intensity is measured using an avalanche photodiode (Thorlabs, APD130A2/M).

## Neural network for waveform recognition

The GAF-ResNet is trained using a graphics processing unit (NVIDIA, GeForce RTX 3070). Each one-dimensional waveform sequence used for training contained 198 samples and a cycle of 2500 points. A waveform is transformed to a 224 × 224 × 3 matrix after the GAF

waveform transformation. After a series of convolution and pooling processes, 256 features are extracted from the matrix and sent to the full-connect layer to calculate one of the 256 categories.

## Data availability

The data supporting the findings of this study are available within the article and its Supplementary Information. Additional data are available from the corresponding author upon request.

## Code availability

The code used in the current study are available from the corresponding author on request.

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

## Acknowledgements

This research was supported by the National Key R&D Program of China under Grant No. 2021YFB3600400, the Mindu Innovation Laboratory Project under Grant No. 2020ZZ113, and National Research Foundation (NRF) of Korea under grants Nos. 2022R1I1A4053429 and 2018R1A5A7025522.

## Author contributions

C.W., Z.L., T.G., and T.W.K. proposed the idea, conceived the experiment, and supervised the research. K.W. constructed the experiment, and acquired and analyzed the data. Y.L. constructed the software and the hardware systems and analyzed the data. W.L. performed the numerical simulations, and acquired and analyzed the data. J.L., H.S., R.C., and J.H.P. wrote the manuscript. X.Z and Y.Z. discussed and analyzed the results. All authors discussed the results and commented on the manuscript.

## Competing interests

The authors declare no competing interests.
