## [Peer Review File · Nature Communications]

REVIEWER COMMENTS

Reviewer #1 (Remarks to the Author):

In this manuscript, the authors demonstrated a novel optoelectronic device based on a Nano-LED with history-dependent luminescence. Based on the device, the conversion of electrical signals to memory-electroluminescence (Mem-EL) spikes is achieved, which mimics the generation of multiple action-potentials and their combinations in a bio-inspired afferent nerve.

The authors used the software and hardware platforms to build a position recognition into the bio-inspired afferent nerve. Moreover, a novel neural network named GAF-ResNet is used to recognize these Mem-EL spikes to mimic the human brain's response to tactile perceptions, achieving a high recognition accuracy.

The idea of this manuscript is novel, interesting and meaningful undoubtedly. I believe that the presented results are solid and an important step in the development of practical mimicking biological afferent nerves and artificial perception systems. Accordingly, this manuscript is suitable for publishing in Nature Communications. The authors are encouraged to address the following issues:

1. The experimental and simulation results show that the luminescence of the nanoLED seems to occur only in the positive half-cycle of the drive signal. What is the role of negative half-period voltage? Is it possible to ignore negative half-cycle voltage?
2. In my opinion, in Figure 4d, due to the discharge effect of the capacitor caused by voltage drop, the number of carriers accumulated on the electrode should decrease. Although there are remaining carriers for the last radiative recombination, the number of carriers should not be more than that in Figure 4c. The author should explain it.
3. As shown in Figures 3e and 3f, the depletion regions of n-GaN and p-GaN show different lengths. The authors should give a reasonable explanation.
4. What is the meaning of "history-dependent luminescence" in Conclusion, and is it similar to memory-electroluminescence (Mem-EL)? If so, the expression should be consistent, otherwise the author should explain the meaning of "history-dependent luminescence" in the manuscript.
5. The memory characteristics of proposed nanoLED are different from traditional memory electronic devices, such as resistive switching device. In order to avoid misunderstanding, it is suggested that the author provide more discussion about it.
6. In the demonstration of the wavelength division multiplexing of the devices, what are the central wavelength of the two filters respectively? And what are the central wavelength of the two device. It is

well known that light will be attenuated when passing through the filter, how does this affect the experiment? such as the recognition of the signal.

7. What is the meaning of the abbreviation “Ce” and “Ch” in Figure 3, respectively. Besides, there are some typos and grammar mistakes in this manuscript, the English should be carefully polished in this manuscript.

Reviewer #2 (Remarks to the Author):

In the manuscript NCOMMS 23-59165 authored by Wang et al., a Nano-LEDs device with memory-electroluminescence (Mem-EL) is demonstrated. Employing this device, electrical signals can be transferred to history-dependent optical signals, which mimics the generation of multiple action-potentials and their combinations in a bio-inspired afferent nerve. The authors combine simulation and experimental results to demonstrate the reasons for the Mem-EL characteristics of the devices and propose a reasonable carrier transport model. Moreover, the wavelength-division multiplexing of the device is demonstrated based on blue and green LEDs, as well as the advantages of the method in mimicking the human perceptual system.

The idea of this design is very progressive, particularly the concept of using Mem-EL optical signals as the transmitted signal in bio-inspired afferent nerve. Due to the special structure of nanoLED devices, the Mem-EL characteristics generated by different Vcombine is almost uniquely, which provides a rich coded space for transmitting optical signals. This innovative idea not only offers promising insights into artificial perception systems but also opens up avenues to study novel electro-optical conversion devices.

For these reasons, I am happy to recommend publication of the work in Nature Communications. However, I have some concerns related to the manuscript, which are outlined below.

- In Introduction, the author wrote "...can be used to transfer electrical signals to history-dependent optical signals and to combine multiple electrical signals into a single light fiber." Why the multiple electrical signals can be combined into a single light fiber?
- The history-dependent EL of the device are is the most important characteristics of the device, which provides the space for optical signal coding. Therefore, it should be clearly described in manuscript.
- The thickness of the sapphire substrate in the actual device structure should be provided. Does it differ from the thickness in the simulation? What factors did the authors consider in determining the Al₂O₃ thickness in simulation?
- In Fig. 3b, the authors used "Forward bias" and "Reversed bias". From my understanding, such a device with capacitive structure, the bias applied to the device should be more dependent on the rate of change of the electric field, and the authors should revise them to avoid misunderstanding.
- The manuscript needs some corrections. Such as

- the overlap between letters and lines in Figure 3d
- The formatting of quotation marks in the caption of Figure 7a
- Please define Ce and Ch in the caption of Figure 3
- Please delete extra commas in the right panel of Figure 5a
- In Figure 7c, The Loss's legend should be corrected
- "...transmitted to a convolutional neural network of recognition..." should be revised to "... for recognition..."

List of changes and responses to reviewers' comments

(Comments in italics, responses in blue, and revisions in yellow highlight)

Reviewers' comments:

Reviewer #1 (Remarks to the Author):

In this manuscript, the authors demonstrated a novel optoelectronic device based on a Nano-LED with history-dependent luminescence. Based on the device, the conversion of electrical signals to memory-electroluminescence (Mem-EL) spikes is achieved, which mimics the generation of multiple action-potentials and their combinations in a bio-inspired afferent nerve.

The authors used the software and hardware platforms to build a position recognition into the bio-inspired afferent nerve. Moreover, a novel neural network named GAF-ResNet is used to recognize these Mem-EL spikes to mimic the human brain's response to tactile perceptions, achieving a high recognition accuracy.

The idea of this manuscript is novel, interesting and meaningful undoubtedly. I believe that the presented results are solid and an important step in the development of practical mimicking biological afferent nerves and artificial perception systems. Accordingly, this manuscript is suitable for publishing in Nature Communications. The authors are encouraged to address the following issues:

Response: We thank the reviewer for carefully reading our manuscript and giving these valuable and positive comments. We have made detailed modifications and additions to the manuscript. Below are our point-by-point responses to your comments.

Comment 1: *The experimental and simulation results show that the luminescence of the nanoLED seems to occur only in the positive half-cycle of the drive signal. What is the role of negative half-period voltage? Is it possible to ignore negative half-cycle voltage?*

Response: We thank the reviewer for this insightful question. The negative half-cycle voltage cannot be ignored, and its role is twofold: first, to release the charge accumulated by the

previous positive half-cycle voltage, allowing the internal carriers of the LED to be restored to the initial state and second, to accumulate sufficient carriers for the radiative recombination in the next positive half-cycle. The experimental and simulation results show that the luminescence occurs only in the positive half-cycle of the drive signal, which is determined by the structure of the pn-junction. An applied electric field acting on the pn-junction is generated under the positive half-cycle of the AC drive signal. When this applied field is larger than the LED turn-on threshold, the carriers inside the Nano-LED are driven into the MQWs to generate radiative recombination, which is equivalent to the forward bias of the pn-junction. However, due to the existence of the insulator, the EL would stop because there are no externally injected carriers for replenishment. The electrons/holes are accumulated at the p-GaN/insulator interface and the n-GaN/insulator interface, respectively. When driven by the negative half-cycle of the AC drive signal, a reverse electric field acting on the pn-junction is generated. The reverse electric field, together with the build-in field, will release the charge accumulated by the previous positive half-cycle voltage, allowing the internal carriers of the LED to be restored to the initial state or sufficient carriers to accumulate for the radiative recombination in the next positive half-cycle.

To further verify the above process, we used the FEA method to simulate a 2D-modeled Nano-LED working in the non-carrier injection mode. The redistributions of the carrier concentrations in the positive and the negative half-cycles are shown in Fig. S6. During the positive half-cycle, the voltage reaches its maximum (Fig. S6a), and the holes in the p-region and the electrons in the n-region are driven into the MQWs under the external electric field. The electron-hole concentrations in the MQW are high, so radiative recombination occurs. An approximately 50-nm-thick region at the top of the p-GaN is depleted, resulting in a very low hole concentration, as shown in Fig. S6b. Similarly, due to the insulating layer between the n-GaN and the external electrodes, a depletion region also exists at the bottom of the n-GaN. Therefore, the electron concentration decreases at the bottom of n-GaN, as shown in Fig. S6c.

At the moment the negative half-cycle voltage reaches its minimum (Fig. S6d), the depletion regions at the tops and the bottoms of the n-GaN disappear, and the hole concentration in the p-region, and the electron concentration in the n-region recover to higher values. Moreover, the carrier concentrations in the MQWs decrease without radiative

recombination, as shown in Figs. S6e and S6f. The carrier concentrations along the dashed lines are provided to show the details. As shown in Fig. S6g, when the voltage reaches its maximum, both the p- and the n-regions show depleted states, and the carrier concentrations in the MQWs reach high levels. However, when the voltage reaches its minimum, the carrier concentrations in the MQWs are insufficient for radiation recombination, and sufficient carriers accumulate in the p- and the n-regions for radiative recombination to occur during the next positive half-cycle.

According to the above discussion, the applied negative half-cycle voltage causes the charge accumulated during the previous positive half-cycle voltage to be released, allowing the internal carriers of the LED to be restored to their initial states or sufficient carriers to accumulate for radiative recombination during the next positive half-cycle. Therefore, the negative half-cycle voltage cannot be ignored.

Figure S6. Finite element analysis (FEA) of the 2D modeled Nano-LED. (a) EL is generated in the positive half cycle of the sinusoidal voltage. Red point: the moment when the sinusoidal voltage signal reaches its maximum. (b) Hole and (c) electron concentration redistributions at that moment in Fig. S6a. (d) No EL is generated in the negative half cycle of the sinusoidal

voltage. Blue point: the moment when the sinusoidal signal voltage reaches its minimum. (e) Hole and (f) electron concentration redistributions at the moment in Fig. S6d. (g) Carrier concentrations along the dotted lines in Figs. S6b and S6e. (h) Carrier concentrations along the dotted lines in Figs. S6c and S6f.

According to the reviewer's valuable comments, we have added the following discussion to the revised manuscript: "The experimental and simulation results show that the luminescence occurs only in the positive half-cycle of the drive signal, which is determined by the structure of the pn-junction. An applied electric field acting on the pn-junction is generated under the positive half-cycle of the AC drive signal. When this applied field is larger than the LED turn-on threshold, the carriers inside the Nano-LED are driven into the MQWs to generate radiative recombination, which is equivalent to the forward bias of the pn-junction. However, due to the existence of the insulator, the EL will stop because there are no externally injected carriers. The electrons/holes are accumulated at the p-GaN/insulator interface and the n-GaN/insulator interface, respectively. When driven by the negative half-cycle of the AC drive signal, a reverse electric field acting on the pn-junction is generated. The reverse electric field, together with the in-built field will release the charge accumulated by the previous positive half-cycle voltage, allowing the internal carriers of the LED to be restored to their initial states or sufficient carriers to be accumulated for the radiative recombination in the next positive half-cycle (Fig. S6 and Text S1)." (line 2, paragraph 1, page 11)

"Moreover, we have added the following description to Supplementary Information: "To further verify the above process, we used the FEA method to simulate a 2D-modeled Nano-LED working in the non-carrier injection mode. The redistributions of the carrier concentrations in the positive and the negative half-cycles are shown in Fig. S6. During the positive half-cycle, the voltage reaches its maximum (Fig. S6a), and the holes in the p-region and the electrons in the n-region are driven into the MQWs under the external electric field. The electron-hole concentrations in the MQW are high, so radiative recombination occurs. An approximately 50-nm-thick region at the top of the p-GaN is depleted, resulting in a very low hole concentration, as shown in Fig. S6b. Similarly, due to the insulating layer between the n-GaN and the external electrodes, a depletion region also exists at the bottom of the n-GaN. Therefore, the electron concentration decreases at the bottom of n-GaN, as shown in Fig. S6c.

At the moment the negative half-cycle voltage reaches its minimum (Fig. S6d), the depletion regions at the top of p-GaN and the bottom of n-GaN disappear, and the hole concentration in the p-region and the electron concentration in the n-region recover to higher values. Moreover, the carrier concentrations in the MQWs decrease without radiative recombination, as shown in Figs. S6e and S6f. The carrier concentrations along the dashed lines are provided to show the details. As Fig. S6g shows, when the voltage reaches its maximum, both the p- and the n-regions show depleted states, and the carrier concentrations in the MQWs reach high levels. However, when the voltage reaches its minimum, the carrier concentrations in the MQWs are insufficient for radiation recombination to occur, and sufficient carriers accumulate in the p- and the n-regions for radiative combination to occur during the next positive half-cycle

According to the above discussion, the applied negative half-cycle voltage causes the charge accumulated during the previous positive half-cycle voltage to be released, allowing the internal carriers of the LED to be restored to their initial states or sufficient carriers to accumulate for radiative recombination during the next positive half-cycle. Therefore, the negative half-cycle voltage cannot be ignored.” (Text S1 in Supplementary Information)

Comment 2: In my opinion, in Figure 4d, due to the discharge effect of the capacitor caused by voltage drop, the number of carriers accumulated on the electrode should decrease. Although there are remaining carriers for the last radiative recombination, the number of carriers should not be more than that in Figure 4c. The author should explain it.

Response: We thank the reviewer for carefully reading our manuscript and pointing this out. As the reviewer mentioned, the discharge effect of the capacitor should cause the number of carriers accumulated on the electrode to decrease.

Therefore, we have revised the Fig. 4 to avoid ambiguity and misunderstanding and have added the following discussion to the revised manuscript: “When the voltage decreases (at this point the applied voltage is V_2 , and $V_2 = V_1$), the induced electric field is greater than the applied electric field. Therefore, the carriers accumulated at the electrode/insulator interface decrease, leading electrons and holes to move to the n-GaN and the p-GaN, respectively. However, the luminescence does not stop immediately because the remaining carriers in the MQWs can still

be used for radiation recombination (Fig. 4d). Therefore, the number of carriers (Q_3) used for radiation recombination is smaller than Q_1 , and the EL intensity is smaller than that in Fig. 4b, leading to the hysteretic EL intensity-voltage loop.” (line 1, paragraph 3, page 15)

Fig. 4 | Carrier transport model of the Nano-LED. **a** Initial state of the device. The number of holes or electrons for radiative recombination in a V_{combine} period is defined as Q . **b** Radiative recombinations of holes and electrons during increases in V_1 . The number of electrons or holes consumed at this stage is defined as Q_1 . **c** Further radiative recombination occurs when the voltage reaches its maximum value. The number of electrons or holes consumed at this stage is defined as Q_2 . **d** Final radiative recombination when the voltage decreases to V_2 ($V_2 = V_1$). The number of electrons or holes consumed at this stage is defined as Q_3 ($Q_3 < Q_1$), $Q = Q_1 + Q_2 + Q_3$.

Comment 3: As shown in Figures 3e and 3f, the depletion regions of n-GaN and p-GaN show different lengths. the authors should give a reasonable explanation.

Response: We thank the reviewer for this valuable comment. The length of the depletion region mainly depends on the doping concentration of GaN. In the simulation model, the doping concentration of holes in the p-region is smaller than the doping concentration of electrons in

the n-region (Table S1). The numbers of electrons and holes used for radiative recombination are assumed to be almost equal when driven by an external electric field. Therefore, the p-region, which has a low hole concentration, requires a longer depletion region to provide an equivalent number of holes as electrons.

We set the doping concentration of electrons in the n-region to be the same as that of holes in the p-region. The simulation results are shown below, and the length of the depletion region is almost the same for n-GaN and p-GaN under this condition.

Figure S11. (a) Electron concentration redistribution in n-GaN and (b) hole concentration redistribution in p-GaN for the same doping concentrations.

Accordingly, we have added the following description to the revised manuscript: “The length of the depletion region mainly depends on the doping concentration of GaN (Fig. S11). In the simulation model, the doping concentration of holes in the p-region is smaller than the doping concentration of electrons in the n-region (Table S1). Therefore, the p-region, which has a low hole concentration, requires a longer depletion region to provide an equivalent number of holes as electrons.” (line 6, paragraph 1, page 13)

Comment 4: *What is the meaning of “history-dependent luminescence” in Conclusion, and is it similar to memory-electroluminescence (Mem-EL)? If so, the expression should be consistent, otherwise the author should explain the meaning of “history-dependent luminescence” in the manuscript.*

Response: We thank the reviewer for this valuable comment. We are sorry that that our descriptions made it difficult for the reviewer to understand. In this work, the history-dependent

luminescence characteristic is defined as that the current luminescence state is highly dependent on the luminescence history. Therefore, as long as the EL intensity of the previous moment is different, the light signal is different even though the amplitude of the currently applied voltage is the same. In other words, the device is capable of memorizing the luminescent state of the previous moment. Therefore, we define the history-dependent luminescence characteristic as memory electroluminescence and have added detailed descriptions of the history-dependent luminescence in the revised manuscript.

To further demonstrate this history-dependent luminescence characteristic, we used two consecutive square signals (the voltages are defined as V_{s1} and V_{s2}) to drive the Nano-LED, as shown in Figs. 2g and 2h. When $V_{s1} = 3$ V and $V_{s2} = 6$ V, the first EL intensity is 0.44, and the second EL intensity is 0.21. However, when V_{s1} is increased to 4 V and V_{s2} is kept constant at 6 V, the first EL intensity increases to 0.57, and the second EL intensity decreases to 0.08. Therefore, the current EL intensity is not necessarily determined by the current voltage but is influenced by the previous EL intensity. In other words, the current EL intensity greatly depends on the historical EL intensity, which is the characteristic of history-dependent luminescence for the device.

Fig. 2 | g EL spikes generated by a Nano-LED driven by two consecutive square signals ($V_{s1} = 3$ V and $V_{s2} = 6$ V) and **h** by another two consecutive square signals ($V_{s1} = 4$ V and $V_{s2} = 6$ V).

Accordingly, we have added the following description to the revised manuscript: “The history-dependent luminescence characteristic is defined as that the current luminescence state is highly dependent on the luminescence history. Therefore, as long as the EL intensity of the previous moment is different, the light signal is different even through the amplitude of the currently applied voltage is the same. In other words, the device is capable of memorizing the luminescent state of the previous moment.” (line 4, paragraph 2, page 4)

“To further demonstrate this history-dependent luminescence characteristic, we used two

consecutive square signals (the voltages are defined as V_{s1} and V_{s2}) to drive the Nano-LED, as shown in Figs. 2g and 2h. When $V_{s1} = 3$ V and $V_{s2} = 6$ V, the first EL intensity is 0.44, and the second EL intensity is 0.21. However, when V_{s1} is increased to 4 V and V_{s2} is kept constant at 6 V, the first EL intensity increases to 0.57, and the second EL intensity decreases to 0.08. Therefore, the current EL intensity is not necessarily determined by the current voltage but is influenced by the previous EL intensity. In other words, the current EL intensity greatly depends on the historical EL intensity, which is the history-dependent luminescence characteristic of the device.” (line 1, paragraph 2, page 8)

Comment 5: The memory characteristics of proposed nanoLED are different from traditional memory electronic devices, such as resistive switching device. In order to avoid misunderstanding, it is suggested that the author provide more discussion about it.

Response: Thanks for carefully reading our manuscript and pointing this out. We fully agree with the reviewer that although the memory characteristics of the proposed Nano-LED are different from those of traditional memory electronic devices, they have similar output performances. As for a memristive device, the current-voltage curve shows a hysteresis-loop characteristic. To illustrate history luminescence or memory electroluminescence more clearly, we further provide the EL intensity-voltage characteristics driven by AC voltages with different amplitudes, as shown in Fig. S4. The lower the applied drive voltage is, the lower the brightness and the smaller the opening of the hysteresis loop are. As the amplitude is increased, the EL intensity increases, which produces a larger hysteresis-loop opening, as shown in Figs. S4a-f. Therefore, the hysteretic EL intensity-voltage curve is similar to the hysteretic current-voltage curve in memristive devices that have history-dependent characteristics. Therefore, we tend to say our device has history-dependent luminescence.

According to the suggestion, more discussion is provided in the revised manuscript: “As is well known, for a memristive device, the current-voltage curve shows a hysteresis-loop characteristic³¹⁻³³. Although the memory characteristics of the proposed Nano-LED are different from those of memristive devices, they have similar output performances. To illustrate history luminescence or memory electroluminescence more clearly, we further provide the EL intensity-voltage characteristics driven by AC voltages with different amplitudes (Fig. S4). The

lower the applied drive voltage is, the lower the brightness and the smaller the opening of the hysteresis loop are. As the amplitude is increased, the EL intensity increases, which produces a larger hysteresis-loop opening. Therefore, the hysteretic EL intensity-voltage curve is similar to the hysteretic current-voltage curve in memristive devices that have history-dependent characteristics. Therefore, we tend to say our device has history-dependent luminescence.”

(line 5, paragraph 1, page 8)

Figure S4. EL intensity-voltage relationship with increasing amplitude of the AC voltage. The amplitudes of the voltages applied to the device are (a) 4.65 V, (b) 4.85 V, (c) 5.30 V, (d) 5.70 V, (e) 6.15 V, and (f) 6.75.

Comment 6: *In the demonstration of the wavelength division multiplexing of the devices, what are the central wavelength of the two filters respectively? And what are the central wavelength of the two devcie. It is well known that light will be attenuated when passing through the filter, how does this affect the experiment? such as the recognition of the signal.*

Response: We thank the reviewer for these valuable comments. The center wavelengths of the two devices are 451.4 nm and 519.8 nm, respectively. To ensure that as much light as possible can pass through the filters, we chose two filters with center wavelengths of 450 nm and 520 nm, respectively. We fully agree with the reviewer that light is attenuated as it passes through the filter. However, as long as enough light passes through the filter, the recognition of the light signal will not be affected. Worth noting is that, in practical applications, the light signals pass

through the filters before training and recognition. Therefore, sufficient light passes through the filters due to their having 88% transparency, so the light signal is sufficient for training and recognition of the signals, and does not affect the experimental process.

Figure S16. The spectra of (a) the blue and (b) the green device.

Accordingly, we have added the following description to the revised manuscript: “The center wavelengths of the two devices are 451.4 nm and 519.8 nm, respectively (Fig. S16). To ensure that as much light as possible can pass through the filters, we chose two filters with center wavelengths of 450 nm and 520 nm, respectively. Worth noting is that, in practical applications, the light signals pass through the filters before training and recognition. Although the light is attenuated as it passes through the filter, as long as enough light passes through the filter, the recognition of the light signal will not be affected. Therefore, sufficient light passes through the filters due to their having 88% transparency, so the light signal is sufficient for training and recognition, and does not affect the experimental process.” (line 1, paragraph 1, page 20)

Comment 7: What is the meaning of the abbreviation “Ce” and “Ch” in Figure 3, respectively. Besides, there are some typos and grammar mistakes in this manuscript, the English should be carefully polished in this manuscript.

Response: Thanks for carefully reading our manuscript and pointing this out. “ C_e ” and “ C_h ” are the abbreviations for electron concentration and hole concentration, respectively. We have defined ‘ C_e ’ and ‘ C_h ’ in the revised manuscript.

Accordingly, we have added the following description to the revised manuscript: “The distributions of the electron and the hole concentrations (C_e and C_h) in the MQWs during a period of V_{combine} are presented in Figs. 3b and 3c.” (line 2, paragraph 1, page 11)

Moreover, we have carefully checked the manuscript and revised the typos and grammar mistakes.

Reviewer #2 (Remarks to the Author):

In the manuscript NCOMMS 23-59165 authored by Wang et al., a Nano-LEDs device with memory-electroluminescence (Mem-EL) is demonstrated. Employing this device, electrical signals can be transferred to history-dependent optical signals, which mimics the generation of multiple action-potentials and their combinations in a bio-inspired afferent nerve. The authors combine simulation and experimental results to demonstrate the reasons for the Mem-EL characteristics of the devices and propose a reasonable carrier transport model. Moreover, the wavelength-division multiplexing of the device is demonstrated based on blue and green LEDs, as well as the advantages of the method in mimicking the human perceptual system.

The idea of this design is very progressive, particularly the concept of using Mem-EL optical signals as the transmitted signal in bio-inspired afferent nerve. Due to the special structure of nanoLED devices, the Mem-EL characteristics generated by different V_{combine} is almost uniquely, which provides a rich coded space for transmitting optical signals. This innovative idea not only offers promising insights into artificial perception systems but also opens up avenues to study novel electro-optical conversion devices.

For these reasons, I am happy to recommend publication of the work in Nature Communications. However, I have some concerns related to the manuscript, which are outlined below.

Response: We appreciate your careful review our manuscript and positive feedback. Your

accurate summary has raised constructive questions. We have made detailed modifications and additions according to those comments. Below are our point-by-point responses to your comments.

***Comment 1:** In Introduction, the author wrote "...can be used to transfer electrical signals to history-dependent optical signals and to combine multiple electrical signals into a single light fiber." Why the multiple electrical signals can be combined into a single light fiber?*

Response: Thanks for carefully reading our manuscript and pointing this out. We are sorry that that our confusing descriptions made it difficult for the reviewer to understand. What we want to say is that multiple electrical signals can be transferred to history-dependent optical signals so that multiple optical signals can then be transported into a single light fiber. Therefore, we have modified the description to make it more readable.

Accordingly, we have added the following description to the revised manuscript: "In this work, we demonstrate that a nanoscale light-emitting diode (Nano-LED) operating in the non-carrier injection mode with memory-electroluminescence (Mem-EL) behavior can be used to transfer electrical signals to history-dependent optical signals. Those multiple optical signals can then be transmitted into a single light fiber." (line 1, paragraph 2, page 4)

***Comment 2:** The history-dependent EL of the device are is the most important characteristics of the device, which provides the space for optical signal coding. Therefore, it should be clearly described in manuscript.*

Response: We thank the reviewer for this valuable comment. In this work, the history-dependent luminescence characteristic is defined as that the current luminescence state is highly dependent on the luminescence history. Therefore, as long as the EL intensity of the previous moment is different, the light signal is different even though the amplitude of the currently applied voltage is the same. Therefore, we added detailed descriptions of the history-dependent luminescence in the revised manuscript.

To further demonstrate this history-dependent luminescence characteristic, two consecutive square signals (the voltages are defined as V_{s1} and V_{s2}) are used to drive the Nano-LED, as shown in Figs. 2g and 2h. When $V_{s1} = 3$ V and $V_{s2} = 6$ V, the first EL intensity is 0.44,

and the second EL intensity is 0.21. However, when V_{s1} is increased to 4 V, and V_{s2} is kept constant at 6 V, the first EL intensity increases to 0.57, and the second EL intensity decreases to 0.08. Therefore, the current EL intensity is not necessarily determined by the current voltage but is influenced by the previous EL intensity. In other words, the current EL intensity greatly depends on the historical EL intensity, which is the history-dependent luminescence characteristic of the device.

Fig. 2 | g EL spikes generated by a Nano-LED driven by two consecutive square signals ($V_{s1} = 3$ V and $V_{s2} = 6$ V) and **h** by another two consecutive square signals ($V_{s1} = 4$ V and $V_{s2} = 6$ V).

In other words, the device is capable of memorizing the luminescent state of the previous moment. Therefore, we define the history-dependent luminescence characteristic as memory electroluminescence. Although the memory characteristics of the proposed Nano-LED are different from those of traditional memory electronic devices, they have similar output performances. As for memristive device, the current-voltage curve shows a hysteresis-loop characteristic. To illustrate the history luminescence or the memory electroluminescence more clearly, we further provide the EL intensity-voltage characteristics driven by AC voltages with different amplitudes, as shown in Fig. S4. The lower the applied drive voltage is, the lower the brightness and the smaller the opening of hysteresis loop are. As the amplitude is increased, the EL intensity increases, which produces a larger hysteresis-loop opening, as shown in Figs. S4a-f. Therefore, the hysteretic EL intensity-voltage curve is similar to the hysteretic current-voltage curve in memristive devices that have history-dependent characteristics. Therefore, we tend to say our device has history-dependent luminescence.

Figure S4. EL intensity-voltage relationship with increasing amplitude of the AC voltage. The amplitudes of the voltages applied to the device are (a) 4.65 V, (b) 4.85 V, (c) 5.30 V, (d) 5.70 V, (e) 6.15 V, and (f) 6.75.

Accordingly, we have added the following description to the revised manuscript: “The history-dependent luminescence characteristic is defined as that the current luminescence state is highly dependent on the luminescence history. Therefore, as long as the EL intensity of the previous moment is different, the light signal is different even though the amplitude of the currently applied voltage is the same. In other words, the device is capable of memorizing the luminescent state of the previous moment.” (line 5, paragraph 2, page 4)

“To further demonstrate this history-dependent luminescence characteristic, we used two consecutive square signals (the voltages are defined as V_{s1} and V_{s2}) to drive the Nano-LED, as shown in Figs. 2g and 2h. When $V_{s1} = 3$ V and $V_{s2} = 6$ V, the first EL intensity is 0.44, and the second EL intensity is 0.21. However, when V_{s1} is increased to 4 V and V_{s2} is kept constant at 6 V, the first EL intensity increases to 0.57, and the second EL intensity decreases to 0.08. Therefore, the current EL intensity is not necessarily determined by the current voltage but is influenced by the previous EL intensity. In other words, the current EL intensity greatly depends on the historical EL intensity, which is the history-dependent luminescence characteristic of the device.” (line 1, paragraph 2, page 8)

“As is well known, for a memristive device, the current-voltage curve shows a hysteresis-

loop characteristic³¹⁻³³. Although the memory characteristics of the proposed Nano-LED are different from those of memristive devices, they have similar output performances. To illustrate history luminescence or memory electroluminescence more clearly, we further provide the EL intensity-voltage characteristics driven by AC voltages with different amplitudes (Fig. S4). The lower the applied drive voltage is, the lower the brightness and the smaller the opening of the hysteresis loop are. As the amplitude is increased, the EL intensity increases, which produces a larger hysteresis-loop opening. Therefore, the hysteretic EL intensity-voltage curve is similar to the hysteretic current-voltage curve in memristive devices that have history-dependent characteristics. Therefore, we tend to say our device has history-dependent luminescence.”

(line 5, paragraph 1, page 8)

Comment 3: The thickness of the sapphire substrate in the actual device structure should be provided. Does it differ from the thickness in the simulation? What factors did the authors consider in determining the Al₂O₃ thickness in simulation?

Response: We thank the reviewer for these insightful questions. As the reviewer mentioned, providing the thickness of the sapphire is crucial for the reader to understand the device's structure and working mechanism. The substrate used for the Nano-LEDs is a commercial sapphire substrate with a thickness of 650 μm .

No doubt, the thickness of the sapphire insulating layer will influence the voltage applied to the device and the EL intensity. However, after considering simulation convenience and computational complexity, we set the thickness of the Al₂O₃ to be 100 nm in the simulation. If the thickness of Al₂O₃ were to be set to the actual thickness (650 μm), it would be much larger than the thickness of GaN ($\sim 2.3 \mu\text{m}$), which would increase the complexity of the mesh division in the simulation and increase the computational cost. Moreover, convergence of the model during the simulation to obtain correct results would be difficult to achieve.

Of note is that the purpose of the simulations is to demonstrate the history-dependent luminescence properties of the devices. Therefore, demonstrating that the mechanism of the restriction process of the induced electric field generated at the previous moment on the current electroluminescence intensity, which is independent of the thickness of Al₂O₃, should be sufficient. We feel that the influence of the thickness of Al₂O₃ on the EL intensity did not affect

the innovation of this work.

Accordingly, we have added the following description to the revised manuscript: “After considering simulation convenience and computational complexity, we set the thickness of the Al_2O_3 to be 100 nm in the simulation. Even though the thickness of the sapphire insulating layer will have an influence on the EL intensity, the purpose of the simulations is to demonstrate the history-dependent luminescence properties of the devices. Therefore, this simulation mode can demonstrate the mechanism behind the generation of history-dependent luminescence.” (line 3, paragraph 1, page 10);

Moreover, we have added the following description to Methods: “A sapphire substrate (650 μm in thickness) is used as the bottom insulating layer, and an ITO film on transparent glass is used as the bottom electrode.” (Methods, page 26)

Comment 4: In Fig. 3b, the authors used "Forward bias" and "Reversed bias". From my understanding, such a device with capacitive structure, the bias applied to the device should be more dependent on the rate of change of the electric field, and the authors should revise them to avoid misunderstanding.

Response: Thanks for carefully reading our manuscript and pointing this out. We fully agree with the reviewer that the bias applied to the device should be more dependent on the rate of change of the electric field. The effect of the change rate of the voltage signal on the charging and discharging of the capacitor is significant, which leads to changes in the operating state of the device. Even if a positive voltage is applied to the device, if the voltage is on the falling edge at that time, the current in the external circuit is reversed, which is completely different from the ‘Forward bias’ in conventional DC operation mode. In a previous manuscript, ‘Forward bias’ meant that the applied electric field was along the direction from p-GaN to n-GaN, and ‘reversed bias’ meant that the applied electric field was along the direction from n-GaN to p-GaN. According to the reviewer’s comment, we have revised ‘Forward bias’ and ‘Reversed bias’ in Fig. 3b to "Positive-half cycle" and "Negative-half cycle", so as to avoid potential confusion.

Accordingly, we have added the following description to the revised manuscript: “Note that the change rate of the voltage signal will lead to changes in the operating state of the device

because of the non-carrier injection mode. Even when a positive voltage is applied to the device, the current in the external circuit is reversed as long as the voltage is on the falling edge. Similarly, when the voltage is on the rising edge of the negative half cycle, the current in the external circuit is positive, which is completely different from the conventional DC mode of operation.” (line 2, paragraph 1, page 14)

“Driven by the positive-half cycle voltage, holes from p-GaN and electrons from n-GaN are transported into the MQWs, and radiative recombination occurs.” (line 1, paragraph 2, page 11)

The revised Fig. 3b, in which "Forward bias" and "Reversed bias" are revised as "Positive-half cycle" and "Negative-half cycle", respectively, is as follows:

Comment 5: The manuscript needs some corrections. Such as

1. the overlap between letters and lines in Figure 3d
2. The formatting of quotation marks in the caption of Figure 7a
3. Please define C_e and C_h in the caption of Figure 3
4. Please delete extra commas in the right panel of Figure 5a
5. In Figure 7c, The Loss's legend should be corrected
6. "...transmitted to a convolutional neural network of recognition..." should be revised to "...for recognition..."

Response: We thank the reviewer for pointing out these mistakes. We have corrected them in the revised version.

1. The relative positions of the letters and lines in Figure 3d are adjusted to avoid overlapping.

The revised Fig. 3d is as follows:

2. We have revised the quotation marks in the caption of Fig. 7a. The revised figure is as follows:

3. C_e and C_h are defined in the caption of Figure 3:

“Fig. 3 | Finite element analysis (FEA) for the Mem-EL. a Structure of the simulated device. **b** Electron and **c** hole concentration (C_e and C_h) redistributions in the MQWs during one drive cycle. The black line is the driving voltage $V_{combine}$, and the blue line is the EL waveform.” (The caption of Figure 3, page 14)

4. We have deleted the extra commas in Figure 5a. The revised figure is as follows:

5. We have corrected the Loss' legend mistakes in Figure 7c. The revised Figure is as follows:

6. We have revised the sentence to avoid grammatical error: “The Mem-EL spikes triggered by distributed sensors are received and transmitted to a convolutional neural network for recognition, which mimics the brain's recognition for spike signals (Fig. 1b).” (line 5, paragraph 1, page 5)

REVIEWERS' COMMENTS

Reviewer #1 (Remarks to the Author):

I think the authors carefully revised the manuscript accordingly. I indeed believe that these results are solid and an important step in the development of artificial perception systems. Therefore, it is suitable for publishing in Nature Communications.

Reviewer #2 (Remarks to the Author):

I have checked the revised version of the paper. The authors have carefully answered the questions raised by the reviewers and revised the manuscript with much necessary data and comments included. The rewritten contents and discussion of the revised manuscript look much better. Therefore, I think this revised manuscript adequately addressed the reviewers' request. It meets the high standards of NC and is suitable for publication now. A very nice work!